# Elevation of Plasma IL-15 and RANTES as Potential Biomarkers of Healing in Chronic Venous Ulcerations: A Pilot Study

**DOI:** 10.3390/biom15030395

**Published:** 2025-03-10

**Authors:** Amanda Beneat, Vikki Rueda, Hardik Patel, Zarina Brune, Barbara Sherry, Andrew Shih, Sally Kaplan, Amit Rao, Annette Lee, Asha Varghese, Alisha Oropallo, Betsy J. Barnes

**Affiliations:** 1Institute of Molecular Medicine, The Feinstein Institutes for Medical Research, Manhasset, NY 11030, USA; abeneat1@northwell.edu (A.B.); hpatel33@northwell.edu (H.P.); zbrune@northwell.edu (Z.B.); bsherry@northwell.edu (B.S.); ashih@northwell.edu (A.S.); atlee1104@gmail.com (A.L.); avarghese@northwell.edu (A.V.); 2Northwell Health Comprehensive Wound Care Healing Center, New Hyde Park, NY 11042, USA; skalplan@northwell.edu (S.K.); arao3@northwell.edu (A.R.); 3Drexel University College of Medicine, Philadelphia, PA 19104, USA; vwr24@drexel.edu; 4Donald and Barbara Zucker School of Medicine at Hofstra/Northwell, Hempstead, NY 11549, USA

**Keywords:** venous ulcer, wound, chronic wound, healing, macrophage, RNA sequencing, IRF5

## Abstract

Chronic wounds present a large burden to our healthcare system and are typically marked by a failure to transition out of the inflammatory phase of wound healing. Venous leg ulcers (VLUs) represent the largest portion of chronic wounds. A pilot study of eleven (11) patients with VLUs seen over a 12-week period was undertaken utilizing RNA sequencing of wound biopsies and plasma cytokine levels to determine if biomarkers could be identified that would distinguish between wounds which heal versus those that do not. Chronic wounds were found to have increased expression of genes relating to epithelial-to-mesenchymal transition (EMT), cartilage and bone formation, and regulation of apical junction. Plasma cytokine levels showed predictive potential for IL-15 and RANTES, which were found to increase over time in patients with healed wounds. Further research is needed to validate these biomarkers as well as additional study of other chronic wound models, such as diabetic foot ulcers (DFUs).

## 1. Introduction

Chronic wounds are estimated to affect approximately 2% of the United States (U.S.) population and pose a significant economic burden [1]. Chronic wounds are common in elderly patients or those with other chronic diseases such as chronic venous stasis, type 2 diabetes, obesity, and vascular disease [1]. Increased costs come from the failure of these wounds to heal, which can lead to recurrent infections, limb loss, and even death. It is estimated that the 5-year mortality of chronic wounds in diabetics contributes to 30.5% of total deaths per year in the U.S. [1,2,3]. As a result, there is a significant need for better diagnosis and treatment of chronic wounds.

Normal wound healing involves 4 stages, (1) hemostasis, (2) inflammation, (3) proliferation, and (4) remodeling, which are tightly regulated by cytokines, chemokines, and growth factors [4,5]. Acute wounds progress through the normal stages of wound healing and show definitive signs of healing within 2–4 weeks, while chronic wounds are defined as those that persist for at least 1–3 months, or even years [1,6]. The inflammatory stage is critical as it initiates the recruitment of immune cells, such as neutrophils, monocytes, and macrophages, to the tissue(s) to facilitate the clearance of cellular debris and combat infectious pathogens [4,7]. However, non-healed (chronic) wounds do not follow the sequential stages of healing and may get “stalled” in one stage, such as inflammation, thus failing to show evidence of healing [8]. Factors that contribute to pathologic inflammation and wound chronicity include poor blood flow, as seen in arterial disease and venous insufficiency, diabetes, increased reactive oxygen species (ROS), and high levels of pro-inflammatory cytokines [8].

Wound healing also involves the transition of epithelial cells to fibroblast-like, migratory mesenchymal cells, which can then enter the wound and begin tissue repair [9]. This process is known as epithelial-to-mesenchymal transition (EMT), and it is a key component of the inflammatory phase of wound healing [9]. In normal wound healing, fibroblasts undergoing EMT migrate to the wound because of platelet and macrophage signaling [4,9]. Once the inflammatory phase completes and the wound enters remodeling, fibroblasts differentiate into contractile myofibroblasts to promote contraction of the wound edges and closure [9]. In chronic wounds, EMT persists due to the continuation of inflammation, which disrupts the normal function of fibroblasts and leads to fibrotic changes in the tissues [9]. Additionally, fibroblasts secrete proteolytic enzymes, such as matrix metalloproteinases (MMPs), which are useful in remodeling and promoting influx of keratinocytes for epithelialization of the wound [4], but in excess can also exacerbate the inflammatory response by cleaving pro-inflammatory cytokines, growth factors, and cell surface receptors, further releasing pro-inflammatory cytokines and perpetuating the inflammatory cycle [4,9,10,11].

Venous leg ulcerations (VLUs) are the most common type of chronic wound and account for 70–90% of all chronic lower extremity ulcerations [6,12]. Caused by chronic venous insufficiency and resultant lower extremity edema, most chronic VLUs fail to heal after 3 months of treatment and can persist for one or more years [6]. VLUs are also at risk of recurrence following effective treatment due to the underlying venous insufficiency [6]. Current practice guidelines recommend early elevation and/or compression of the affected extremity to relieve the edema brought on by venous insufficiency, and treatment of the wound through debridement, or removal of necrotic tissue, debris, and possible biofilms; control of moisture and infection, and maintenance of wound edges to promote epithelialization of the wound [12,13]. For patients with severe venous disease or those who have demonstrated reflux of veins directed towards the wound bed, surgical therapy such as venous ablation may be offered [13]. The incompetent valves seen in chronic venous insufficiency lead to venous hypertension, edema of the affected limb, and disrupted endothelial integrity [14,15]. Further, increased venous pressure results in endothelial fenestrations as well as activating endothelial cells to release chemoattractants for leukocyte recruitment, contributing to inflammation [14]. Over time, limb edema leads to microtears in the skin, which then progress to ulceration [14].

Predicting and identifying non-healing VLUs at initial presentation is essential to reducing overall costs and patient morbidity [16]. One multi-center study found that patients with an increase of 3% or more in wound area over the first 4 weeks were 68% more likely to have a persistent wound at 24 weeks [17]. Meanwhile, another multi-center study created a risk assessment tool that has been able to identify potential non-healing VLUs after 2 weeks or treatment [18]. Identification of biomarkers to predict wound healing has also been studied [19], but there is not a consensus on any biomarkers as an early predictor of wound healing. Such biomarkers may be related to inflammation, EMT, or yet be undetermined. This study aimed to investigate gene expression changes at the wound bed in patients with VLU in comparison with changes in plasma cytokine levels over time. We hypothesized that chronic wounds would have over-expression of genes related to the inflammatory stage of wound healing, indicating a failure to transition out of this phase.

## 2. Materials and Methods

### 2.1. Patients

#### 2.1.1. Ethics Statement

The study adhered to the ethical principles outlined in the Declaration of Helsinki. The medical protocol was reviewed and approved by the Northwell Institutional Review Board (IRB) and Office of the Human Research Protection Program and was classified as minimal risk. Participants were selected according to the IRB protocol #14-328. This was a pilot study and thus consisted of a smaller sample size. Planned enrollment was for 20–25 patients in the initial study, with planned expansion based on a formal power analysis and sample size calculation.

#### 2.1.2. Patient Selection

Patients over 18 years diagnosed with a venous stasis ulcer that persisted after 4 weeks despite conservative management, measuring at least 1–2 cm at the time of enrollment, and who were determined to have adequate vascular perfusion by ankle-brachial index (ABI) > 0.9 and < 1.2, plethysmography, Doppler waveform, or clinical evidence of palpable pedal pulses were considered for this study. Venous stasis ulcers were defined per the Society of Vascular Surgery guidelines as a full thickness defect of the leg or ankle in the setting of venous insufficiency, diagnosed by venous duplex ultrasound [13]. All participants signed the informed consent form to express their willingness to participate. All patients were followed for at least 12 weeks in total or until wounds had healed.

Patients with suspected skin or wound infection or confirmed osteomyelitis, those taking immunosuppressive therapies at the time of the study or with a history of radiation to the ulcerated region, a Body Mass Index (BMI) ≥ 39, or those with an autoimmune or connective tissue disorder were excluded.

### 2.2. Sample Collection

#### 2.2.1. Tissue

Sharp debridement of the ulceration was carried out per the consensus of the Chronic Wound Care Guidelines for foot ulcers and the Society of Vascular Surgery guidelines on management of venous leg ulcers [13,20]. All non-living and/or contaminated tissue was removed such that a healthy-appearing wound bed was visible. Two 3 mm punch biopsies were taken from the ulcer bed, including the wound edge, and preserved in RNALater (Ambion/Applied Biosystems, Oslo, Norway) for gene expression analysis. Wound biopsies were taken at the time of enrollment, which was designated as “week 1” of the study timeline.

#### 2.2.2. Blood and Plasma

Blood was taken at enrollment and weekly visits or until the ulcer had healed (whichever was sooner). The earliest healing timepoint was 10 weeks post-enrollment; therefore, this was determined as the final blood sample for the study. Approximately 4 mL of whole blood was collected from non-fasting patients in an EDTA tube (BD, Franklin Lakes, NJ, USA), and plasma was collected according to standard protocols [21].

#### 2.2.3. Wound Measurements

Wound measurements and photographs were performed according to standard of care [20,22], where the greatest length and width were recorded to determine wound area. Wound healing was determined according to standard guidelines, and a wound with a 100% reduction in wound area for at least two (2) weeks was considered healed [23].

### 2.3. RNA Sequencing and Genomic Analysis

#### 2.3.1. Bulk RNA Sequencing

Wound biopsy tissues were stored in RNALater and sequenced in bulk. Libraries for sequencing were made from total RNA using a stranded mRNA library kit (Illumina, San Diego, CA, USA). Eleven (11) samples were sequenced (5 healed and 6 non-healed) using a high output flow cell on the Illumina NextSeq (Illumina, San Diego, CA, USA). Gene expression between groups was assessed based on log2 fold change. A false detection rate (FDR) analysis was used to adjust *p*-values. Selected genes related to wound healing or inflammation or those relating to plasma cytokines being studied underwent additional comparison between groups. All expression levels were normalized using log2 transformation.

#### 2.3.2. Gene Set Enrichment Analysis

Identification of enriched biological pathways associated with non-healed venous ulcerations was performed using the “GSEA (v4.3.3)” gene set enrichment analysis software package [24]. Gene expression levels were compared between “healed” and “non-healed” groups. Pre-defined gene sets from the Molecular Signatures Database (MSigDB) from the Hallmark collection, which curates all MSigDB sets into a single package to eliminate redundancy [25]. An enrichment score was calculated via a weighted Kolmogorov–Smirnov-like test. Statistical significance was assessed through phenotype-based permutation testing. Enrichment scores were normalized to account for multiple hypothesis testing [24].

### 2.4. Cytokine Analysis

#### 2.4.1. Collection

Plasma was extracted from whole blood samples and saved for bulk analysis. Blood samples were taken at the time of enrollment and weekly thereafter until the wound had healed or the 12-week study end point had been reached. Three patients in the healed group reached their endpoint before the end of the 12-week study period. To adequately match each sample, an endpoint for plasma analysis was chosen to be 10 weeks post-enrollment. Plasma was analyzed for 15 inflammation-related cytokines using Multiplex Elisa kits (MSD, Rockville, MD, USA). Cytokines analyzed were IFNγ, IL-10, IL-8, MCP1, IP10, TNF, IL6, MIG, TARC, RANTES (CCL5), IL-18, IL-2, ITAC, IL-1α and IL-15. This kit was chosen as it included many of the major inflammatory cytokines and chemokines related to wound healing and allowed for the analysis of multiple cytokines/chemokines within a single patient sample.

#### 2.4.2. Analysis of Cytokine Expression over Time and Comparison to mRNA Expression

The average percent change in each group, in aggregate, over 10 weeks was generated to compare trends in cytokine levels between the healed and non-healed cohorts. An additional comparison was performed using patient-specific percent change in cytokine levels.

Correlation analysis was performed between mRNA levels and plasma cytokine levels. The tissue samples for RNA sequencing were taken at the time of enrollment (week 1); therefore, these were compared to week 1 plasma cytokine levels as well as individual mean percentage-change over the 10 weeks. Raw values were normalized using log2 transformation.

#### 2.4.3. Plasma Biomarker Analysis

Mean percent-change between week 1 and week 10 cytokine levels was used for univariate logistical regression to determine likelihood ratios (LR) and odds ratios (OR) for each cytokine studied. Significance for LR and OR was assessed using a *p*-value and 95% confidence interval, respectively. Multivariate logistic regression was performed on any cytokines with significant predictive potential (LR with significant *p*-value and 95% CI).

### 2.5. Statistical Analysis

All statistical analysis was performed using GraphPad Prism version 10.4.0 for Mac OS, GraphPad Software, Boston, MA, USA, www.graphpad.com. Quantitative data are presented as mean ± SEM. An adjusted *p* < 0.05 was considered statistically significant. All data were assessed using an unpaired *t*-test unless otherwise noted. Before the test, graph kurtosis was used to determine normal distribution.

## 3. Results

### 3.1. Patient Population

Twenty-five (25) patients were recruited for this pilot study who met criteria for inclusion. Of these, four (4) met exclusion criteria and thus were not included, with three (3) having A1c > 6.5% (therefore meeting criteria for diabetes) and one (1) having a diagnosis of scleroderma. Two (2) were later withdrawn after developing cellulitis requiring antibiotics and two (2) rescinded consent mid-study. An additional six (6) were lost to follow up or contained missing data at the end timepoint and were therefore excluded from analysis. The cohort consisted of 4 females (36.4%) and 7 (63.6%) males with the average age being 63.8 ± 13.2 years (Figure 1). As per the Guidelines for Treatment of Venous Ulcers [22], six (*n* = 6) were determined to have non-healed wounds and five (*n* = 5) to have healed wounds at the termination of the 12-week study. Of the six (6) non-healed wounds, four (4) were found to have an increase in wound area by >3% (66.67%), while no patients in the healed cohort showed an increase in wound area.

### 3.2. Patient and Wound Characteristics

Patient demographics, as well as smoking status and common comorbidities, is shown inTable 1. There were no significant differences between groups in terms of race, ethnicity, gender, BMI, medical comorbidities, or smoking status. All patients presented with a medial malleolar ulceration (chronic venous ulcer) that had been present for at least 4 weeks despite medical management. All patients were diagnosed with venous insufficiency based on duplex imaging. During our study, all wounds were managed conservatively with compression, dressing changes, and routine wound cleansing and debridement as needed. Surgical interventions were not offered during the period of the study; however, one (1) patient was offered skin grafting after the study had concluded.

Characteristics of patient ulcerations is presented inTable 2. There was no difference seen in the average age of the ulceration at the time of presentation (Table 2). In the unhealed group, the average age was skewed to 39.5 months by a single patient whose wounds had been present for over 10 years; however, the remaining patients had an average wound age of 11.4 (±8.1) months, while the average age in the healed group was 7.8 (±4.0) months. Both groups had similar baseline wound sizes: 11.2 cm^2^ for the unhealed group and 9.37 cm^2^ in the healed group. The average healing time for the healed group was 4.4 months, while wounds in the unhealed group persisted for 1 year or longer.

### 3.3. Wound Biopsy Gene Expression

#### 3.3.1. Most Highly Expressed Genes Are Related to Cell Signaling, Collagen Formation, and Extracellular Matrix Deposition

Wound biopsy samples following debridement were obtained at initial visits (week 1) for transcript expression by RNAseq. The top 20 highly expressed protein-coding mRNA transcripts in the non-healed group were determined after the log2 fold change was calculated between groups (Figure 2). All non-coding RNA strands were excluded from this analysis.

Top genes were found to be associated with cell adhesion; cell growth, signaling, and migration; collagen, bone, and cartilage formation; and neuronal development (Table 3).

#### 3.3.2. Gene Set Enrichment Analysis Reveals Pathways Enriched in Non-Healed Cohort

A gene set enrichment analysis was performed using the Hallmark gene set from the program “GSEA”. Of the 50 gene sets analyzed, six (6) were found to be enriched in the non-healed patient cohort. These were pathways relating to myogenesis, epithelial-to-mesenchymal transition, KRAS signaling, apical junction-related genes, and genes related to coagulation and oxidative phosphorylation (Figure 3a). Interestingly, genes related to inflammatory response and interferon-, TNFα- and TGFβ-signaling were found to be more enriched in the healed cohort (Figure 3b).

#### 3.3.3. In-Depth Analysis of Inflammatory Gene Expression

Next, we took a deeper dive into genes that are known to be related to inflammation and wound healing to determine differential expression between groups (Figure 4).

Transcription levels of all cytokines investigated in plasma studies (as discussed in Section 2.3.1) and those important to healing in VLUs, were assessed, as well as several additional cytokines noted to be important to healing in VLUs; Interleukin 1 beta (IL1b), Transforming Growth Factor beta-1 (TGFb1), and Vascular Endothelial Growth Factor A (VEGFA) [26]. The lowest levels of expression across both groups were seen in *IFNγ* (mean = 2.203 ± 0.646) and *IL2* (mean = 1.304 ± 0.405), while the highest levels of expression were found in *CXCL8* (IL8) (mean = 14.164 ± 1.011) and *VEGFA* (mean 12.814 = ±0.361) (Figure 4a). Only *RANTES* showed a significant increase in gene expression in the non-healed group (*p* = 0.0303) (Figure 4b).

Transcript levels for several matrix metalloproteases (MMPs) were also examined, specifically *MMP1*, *3*, *8*, and *9*, which are known to play a role in wound chronicity [10,11], and *MMP2* and *7*, which have an increased presence in normally healing wounds [11] (Figure 4c). In both groups, *MMPs 1*, *2*, and *3* were most highly expressed (Figure 4d,e). However, there was no difference between healed and non-healed groups at the transcription level (Figure 4d).

Given the role of macrophages in wound healing, we next examined the expression of macrophage-polarizing transcription factors interferon regulatory factor 4 (IRF4) and 5 (IRF5). We found that the non-healed cohort had higher levels of expression of *IRF5,* which drives M1-like macrophage polarization, than the healed cohort (Figure 5). This was found to be significant when normalized expression levels were compared between the groups (*p* = 0.0121, 95% CI 0.2978 to 1.846) (Figure 5a). Interferon regulatory factor 4 (IRF4), which plays a role in M2-like macrophage polarization, appeared to have decreased expression in the non-healed cohort compared to the healed; however, this was not significant. Additionally, when comparing the log fold change between healed and non-healed cohorts, *IRF5* was found to have significant expression in the non-healed group. *IRF4* expression was increased in the healed cohort, although this was not significant (Figure 5c).

### 3.4. Plasma Cytokine Analysis

#### 3.4.1. Differences in Cytokine Levels

Blood was taken from patients at the time of debridement (week 1) and at week 10. An aggregate analysis of cytokine levels was performed using percent change between the two time points, calculated using an average of all patients in each group to determine trends in plasma cytokine levels. Cytokines analyzed were Interferon Gamma (IFNγ), Tumor Necrosis Factor (TNF), IL-1α, IL-2, IL-6, IL-8 (CXCL8), IL-10, IL-15, IL-18, Monocyte Chemoattractant Protein 1 (MCP1 or CCL2), Regulated on Activation, Normal T Cell Expressed and Secreted (RANTES or CCL5), Thymus and Activation Regulated Chemokine (TARC or CCL17), Monokine Induced by Gamma Interferon (MIG or CXCL9), Interferon Gamma-induced Protein 10 (IP10 or CXCL10), and Interferon-inducible T-cell Alpha Chemoattractant (I ITAC or CXCL11).

In the healed cohort, cytokines with a greater than 10% increase included IFNγ, IL-15, IL-18, IL-1α, IL-2, IL-6, IL-8, ITAC, MIG, TARC, and RANTES (Figure 6a). Similarly, in the non-healed group, IFNγ and ITAC had a greater than 10% increase, but unique to this group were increased IP10 and IL-10. Additionally, there were decreases of more than 10% over the 10-week period of IL-1α, IL-2, RANTES, TARC, and TNF in the non-healed group. Notable cytokines from this analysis were RANTES and IFNγ. There was a 59% increase in RANTES plasma levels in the healed cohort and a corresponding 48% decrease in the non-healed cohort (Figure 6a). We also detected a 67% increase in IFNγ plasma levels in healed patients, compared to only a 24% increase in the non-healed group.

The average percent change between both time points for each individual patient was also calculated for comparison. Despite the large difference between groups seen in RANTES when overall percent change was assessed, the difference between individual patient values was not significant (*p* = 0.0786, 95% CI –2.305 to 0.1511) (Figure 6b). Interestingly, the increase in IFNγ levels seen in the healed cohort when comparing overall percent change between groups was not seen when the percent change for individual patients was compared. Instead, an increase in plasma levels in non-healed patients was noted; however, this was not significant (*p* = 0.1233, 95% CI –0.1409 to 0.9943) (Figure 6b). Only IL-15 was found to be significantly different between groups, with an increased level seen in healed patients (Figure 6c); (*p* = 0.0301, 95% CI –0.4708 to –0.03012).

#### 3.4.2. Correlation Analysis of Gene Expression and Cytokine Levels

The level of RNA transcripts from tissue samples taken at week 1 (time of enrollment) were compared to plasma cytokine levels, both from week 1 blood draws and average percent change between week 1 and week 10 plasma levels.

We initially compared RNA expression from wound biopsies to plasma cytokine levels taken at the same timepoint (time of enrollment), to determine if a correlation existed between gene expression and appearance in plasma (Figure 7). There were no significant correlations; however, a strong positive correlation for IL-6 in the non-healed cohort was seen. (r coefficient = 0.8046, *p* = 0.053).

We next performed an analysis of RNA levels at initial debridement in comparison to changes in plasma levels over time using the mean percent change in plasma levels for each patient (Figure 8). In the non-healed cohort, a positive correlation was seen between *RANTES* RNA at initial debridement and changes in plasma RANTES over time (r coefficient = 0.8234, 95% CI 0.03554 to 0.9800, *p* = 0.0440), while a strong negative correlation was seen between *MIG* RNA and changes in plasma over time (r coefficient = −0.8403, 95% CI −0.9821 to −0.09034, *p* = 0.0362). In the healed cohort, a strong positive correlation was seen between *TARC* RNA at initial debridement and changes in plasma TARC over time (r coefficient = 0.9112, 95% CI 0.1473 to 0.9942, *p* = 0.0314).

#### 3.4.3. Predictive Potential of Plasma Biomarkers

A univariate regression analysis for each of the fifteen (15) plasma cytokines, utilizing percent change in plasma levels between week 1 and week 10, determined that only IL-15 and RANTES had significant predictive potential for differentiating between a healing and non-healing wound. IL-15 was found to have a likelihood ratio (LR) of 6.395 (*p* = 0.0114), and an increase in IL-15 over the 10 weeks was associated with an odds ratio (OR) of 2.6 × 10^7^ (95% CI 9.276 to 1.276 × 10^20^). RANTES was found to have an LR of 4.058 (*p* = 0.044) and OR 5.827 (95% CI 1.038 to 273.6). The remaining cytokines (IFNγ, IL-10, IL-8, MCP1, IP10, TNF, IL6, MIG, TARC, IL-18, IL-2, ITAC, and IL-1α) did not have significant predictive value upon univariate analysis.

A multiple logistical regression was performed with both cytokines to determine if this predictive power could be additive; however, confidence intervals were not significant using this model.

## 4. Discussion

In this study, we investigated gene expression differences between chronic VLUs that healed with non-surgical wound care compared to those that did not and probed for potential plasma biomarkers that could lead to the early identification of wounds that may be resistant to conservative therapy alone. Prolonged inflammation leads to failure of wounds to heal [5,27], and current practice guidelines recommend at least 4–6 weeks of conservative management before additional therapies are discussed [13]. The goal of our study was to identify potential biomarkers of wound healing that can be predictive even at the initial presentation of a chronic wound, to better identify and treat patients with chronic VLUs and better stratify patients into more aggressive therapy without the initial 4–6 weeks of management.

When the top 20 protein-coding RNAs were analyzed between groups, we found them to be primarily related to cell adhesion, ECM formation, collagen, cartilage and bone deposition, and neuronal growth. Genes related to chondrogenesis and bone deposition (*OGN*, *EPYC*, *COL11A1*, *SCRG1*) were an interesting finding in the non-healed VLU cohort [28]. Heinz-Lippmann disease, an acquired heterotopic ossification (HO) of soft tissues surrounding VLU, is a rare yet underdiagnosed complication of chronic VLUs [29,30]. HO is most common in traumatic injuries, and the heterotopic bone formation seen in Heinz-Lippmann disease is related to microtrauma at the cellular level [29]. COL11A1 is increased in wound tissues of patients with acquired HO as part of the BMP/TGFβ signaling pathway [31,32]. While the roles of OGN, EPYC, and SCRG1 in Heinz-Lippmann are unclear, upregulation of OGN has been implicated in the heterotopic ossification of the posterior longitudinal ligament [33], and SCRG1 has been shown to play a role in the remodeling and regeneration of mesenchymal cells in bone [34]. Recent case reports have noted the potential for osteomyelitis in the heterotopic bone [30], which can increase inflammation and prevent healing. Thereby making the presence of these genes at initial presentation important for identification of patients who may be at risk for HO and therefore less likely to heal.

Gene set enrichment analysis also revealed increased transcripts of genes related to the regulation of the apical junction (i.e., cell signaling and adhesion), coagulation, and epithelial-to-mesenchymal transition (EMT). EMT, while a normal part of the wound healing process, has been shown to play a role in wound chronicity when dysregulated [35]. EMT, as seen in wound healing, involves several pathways of Wnt signaling [9]. The Wnt-β catenin pathway (or canonical Wnt pathway) is involved inthe regeneration of the interfollicular epidermis and is involved in the regulation and enhancement of the inflammatory process of wound healing [36,37]. This pathway also leads to upregulation of fibroblasts and ECM remodeling factors such as MMPs 2, 7, and 9 [35,36], which leads to successful remodeling of the tissues. In our patient sample, MMPs 7 and 9 were found to have low levels of expression in both groups, whereas MMPs 1–3 were more highly expressed in both groups. This may be due to the roles of MMPs 1–3 at the wound edge, which is where the biopsies had been taken [10]. Additionally, the Wnt/PCP pathway, a non-canonical pathway that induces cell migration and regulation of cell polarity, has also been implicated in wound healing [35]. WISP-2, or WNT-1 Inducible Signaling Pathway Protein 2, has been shown in a model of esophageal cancer to negatively regulate the Wnt/β-catenin pathway and reduce cell migration [38]. This gene was upregulated in our non-healed cohort, suggesting there may also be an aspect of inhibition of Wnt-induced cell migration in wound tissues.

The inflammatory gene set was more enriched in the healed cohort, which was counter to our initial hypothesis. However, this is not surprising given all tissue samples were taken at the initial presentation, prior to starting a regular wound care regimen. Therefore, it would be expected that all wounds might be in an inflammatory state. To understand if markers of chronicity exist to differentiate a wound that may respond to conservative therapy versus those that would not, we attempted to look at some genes that play a role in inflammation.

One such gene was the transcription factor interferon regulatory factor 5 (IRF5) which is an important regulator of macrophage function. IRF5 expression is high in pro-inflammatory M1-like macrophages and low in anti-inflammatory M2-like macrophages, thus, IRF5 is used as a marker of polarization [39,40]. A failure to repress pro-inflammatory mediators such as IRF5 in the wound microenvironment leads to phenotypic switching of macrophage subsets from M2- to M1-like and can result in poor healing [39,41,42]. The counterbalance to IRF5 is the anti-inflammatory IRF4, which has been shown to promote M2-like macrophage polarization [43]. Although the expression in the non-healed cohort was not significantly decreased compared to the healed group, the lower level of expression could result in less M2 polarization, which, coupled with increased *IRF5*, would promote a predominance of inflammatory cells in the wound bed.

MMP expression at an initial presentation was consistent between both groups, with MMPs 1–3 being the most highly expressed. MMP1 and 3 are more active in chronic wounds, and MMP2, often seen in remodeling stages of healing, can also be expressed during inflammation [11,44]. The lack of differentiation between the two groups may be the result of the sampling at only one timepoint, and that expression of these MMPs may likely change over time to favor either a healing or non-healing wound profile.

In addition to transcription differences between healing and non-healing VLUs, we investigated plasma cytokine levels both at an initial presentation, as well as at 10 weeks post enrollment. We attempted to correlate both plasma cytokine levels at initial presentation with RNA expression, as well as a change in plasma cytokine levels over time, to determine if RNA expression could be predictive of healing. We found that comparing RNA expression at initial presentation with changes in plasma cytokine levels over time provided better insight into the identification of healing in VLUs than comparing both at the time of enrollment alone.

Despite initially showing higher gene expression in the non-healed group at initial presentation, RANTES showed a strong correlation between levels of RNA expression in the non-healed cohort and an overall decrease in plasma levels over 10 weeks. RANTES is a chemokine that acts to recruit monocytes, lymphocytes, mast cells, and activated T-cells [45]. It has been observed in several inflammatory diseases that affect the kidney, joints, and bowel, and the inactivation of its receptors has been shown to have protective effects [45,46,47]. In contrast to these studies, RANTES was shown to aid in the resolution of allergic asthma when stimulated with recombinant formulations or by TLR7/8 stimulation with R848 [48]. TLR7/8 agonists such as R848 or Imiquimod have been used topically to affect wound healing [49,50,51,52]; however, these studies showed prolonged re-epithelialization and decreased healing. Imiquimod is an FDA-approved agent for many cutaneous neoplasms due to its ability to induce apoptosis [53]. Further, RANTES has been shown in murine models to be highly expressed by keratinocytes [54], and wound healing assays utilizing human-derived stem cells have shown that RANTES acts as a potent promoter of stem cell migration to aid in wound closure [55]. This contrast suggests that RANTES activation, outside of the TLR pathway, may have a role in healing. Our data show that RANTES increased over time in the plasma samples in the healed cohort and decreased in the non-healed group, further supporting this notion.

When assessed for its predictive potential utilizing the likelihood ratio function of a regression analysis, it was determined that these changes in plasma levels have a significant predictive potential. Coupled with the positive correlation to RNA levels in the non-healed cohort, there exists a potential for identification of patients at risk for non-healing wounds by measuring RANTES in wound tissue at the time of presentation.

Plasma IL-15 levels significantly increased over time in the healed cohort, despite there being no difference between groups at the RNA level at initial enrollment. There also did not appear to be a correlation between plasma IL-15 levels and RNA expression in either group. IL-15 has been shown to play a role in both the innate and adaptive immune systems in response to pathogens [56]. IL-15 is a regulatory cytokine that is produced by a wide expanse of cell types, excluding T cells [57,58]. IL-15 has long been studied for its anti-tumoral properties [59,60,61], and several recombinant forms of IL-15 agonism exist [59,62,63]. One IL-15 agonist was recently approved by the FDA for the treatment of bladder cancer [60]. IL-15 has also been shown to play a regenerative role in a murine model of liver injury [64]. In the skin, it is produced by keratinocytes, and it has also been linked to dendritic epidermal T cells (DETCs), which are required for wound repair [65]. It has also been shown to have a positive effect on keratinocyte migration and anti-apoptotic functionality [66,67,68]. In a murine model of diabetic wounds, reduction in IL-15 led to impaired DETC production and poor wound healing, while low-dose intravenous application of recombinant IL-15 rescued chronic wounds [65,69]. Transgenic mice showing increased expression of IL-15 had similar improvements in wound healing but notably had decreased protection from viral infections, such as HSV-2 [70]. While these studies point to a therapeutic potential for IL-15, a topical formulation would likely have greater clinical translation and avoid sequelae of systemic immunomodulation.

In a similar study where wound biopsies were taken from 71 VLU patients, *IL15* RNA was shown to be elevated in the healed cohort, though this was not significant [66]. This study did not investigate IL-15 levels in plasma of VLU patients, and a review of the literature suggests that many previous biomarker studies focus primarily on wound biopsy and less commonly on wound fluid or blood samples for analysis [26]. The change in plasma levels seen in our study was found to have strong predictive potential following regression analysis and an extremely high odds ratio, suggesting increasing levels of IL-15 is associated with a healing wound. Plasma IL-15 levels may, therefore, be a useful biomarker in the determination of healing wounds, and further study should be considered.

This study has several limitations. The small sample size may lead to an underpowered analysis, although comparable studies with larger sample sizes showed similar results. Additionally, as wound healing is a dynamic process involving a balance between pro-inflammatory and anti-inflammatory states, there may be an overlap with pro-inflammatory cytokines detected in healed wounds and anti-inflammatory biomarkers in non-healed wounds. This underscores the importance of frequent sampling of wounds to better understand the preferred treatment for specific healing phases of each patient and each wound. Thus, sampling of wound tissue only at the time of enrollment in this pilot study poses a limitation, and future studies will, therefore include tissue samples taken at additional time points to determine if gene expression levels change as wounds progress to either complete healing or remain unhealed. A major limitation of our study is the presence of multiple wounds in several patients in both groups, which is not uncommon to patients with VLI in our practice. These additional wounds could result in continued systemic inflammation despite healing of the VLU, which may have confounded our results and led to the lack of differences between some of the inflammatory plasma cytokines. A final limitation was the loss of follow-up from two of the patients in the non-healing group. Although all 11 patients were successfully followed until at least 12 weeks, loss of follow-up prior to healing provides a truncated timeline for healing, which may have contributed to the lack of significance between the healed and non-healed groups when this metric was assessed.

## 5. Conclusions

Non-healing wounds had increased transcript levels of the *RANTES and the* pro-inflammatory transcription factor *IRF5*. However, increases in plasma levels of RANTES and IL-15 in the healed groups were determined to have more value in predicting healing when measured over time.

## Figures and Tables

**Figure 1 biomolecules-15-00395-f001:**
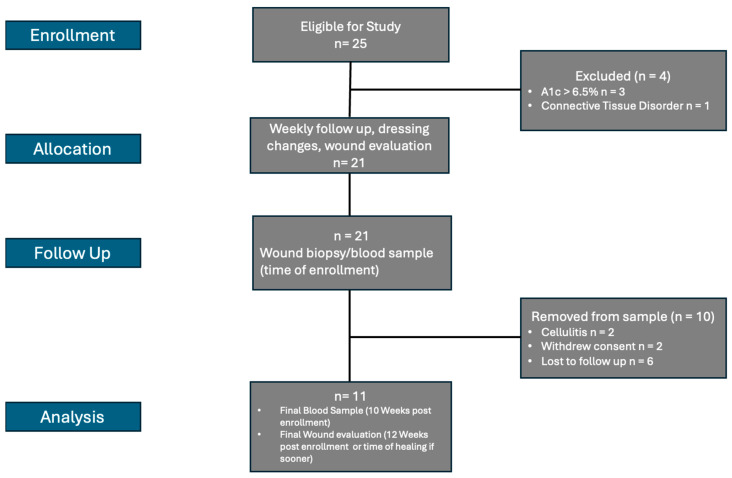
Flow Diagram of Patient Selection.

**Figure 2 biomolecules-15-00395-f002:**
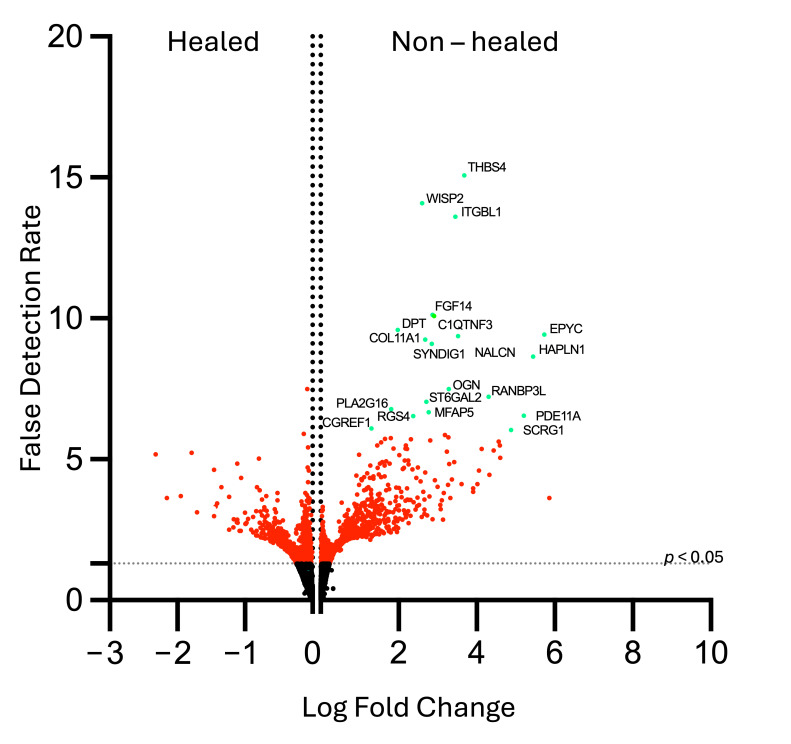
Top 20 protein coding RNAs in non-healed wounds favor cell signaling, collagen and bone formation, and ECM deposition. Tissue taken during routine wound debridement from venous ulcerations at the initial visit was analyzed by bulk RNAseq. At the end of the study, the expression levels were compared using log2 fold change between groups and a significant *p*-value was set to 0.05 (red dots) Twenty protein−coding RNAs with the most significant *p*-value (adjusted using false detection rate), were found to be related to cell adhesion; cell growth, signaling, and migration; collagen, bone, and cartilage formation; myogenesis; and neuronal development.

**Figure 3 biomolecules-15-00395-f003:**
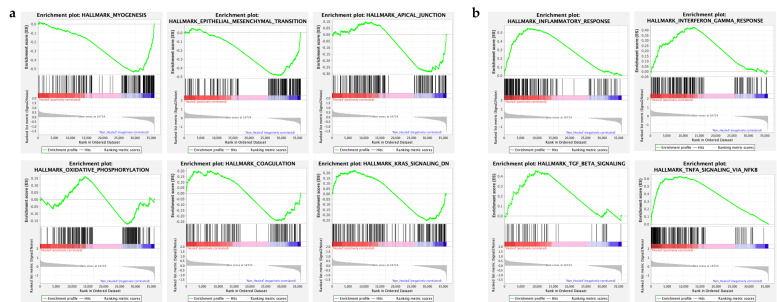
Gene Set Enrichment Analysis for Healed vs. Non-Healed Cohorts. Wound biopsies taken at week 1 were evaluated for RNA expression by bulk RNAseq and GSEA was performed between groups. (**a**) Six gene sets found to be enriched in the non-healed cohort were related to myogenesis, EMT and apical junction, KRAS signaling, and coagulation and oxidative phosphorylation. Myogenesis, EMT, and Apical Junction gene sets appear to have strong peaks only on the non-healed side, whereas oxidative phosphorylation, coagulation, and KRAS signaling appear to have a more equal distribution of genes. However, none were found to be significant when using FDR q-value to adjust for differences in gene set size and multiple hypothesis testing. (**b**) Gene sets enriched in the healed cohort included inflammatory response, interferon gamma response, TGF beta signaling, and TNFA signaling via NF*κ*B. Again, when the FDR q-value was used to adjust raw *p*-values, no significant differences were seen.

**Figure 4 biomolecules-15-00395-f004:**
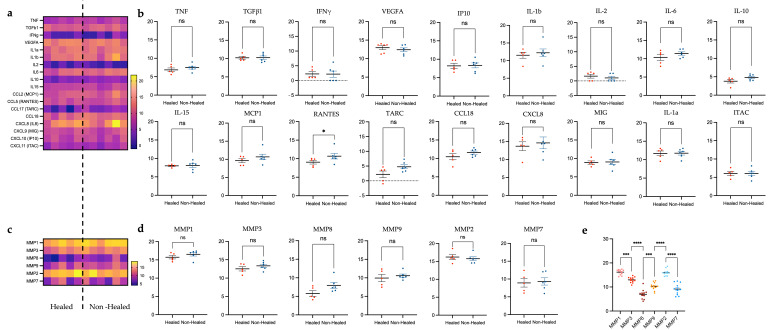
In depth analysis of select inflammatory cytokines, chemokines, and genes of interest from debrided tissue take at week 1. RNA sequencing of tissue taken from all patients during routine wound debridement at week 1. When all inflammatory cytokines or chemokines studied in plasma samples as well as others involved in wound healing (*IL1β*, VEGFA, *TGFb1*, *CCL18*) were analyzed, only *RANTES* showed a significant increase in the non-healed group at week 1 (**a**,**b**). Similarly, matrix-metalloproteases (MMPs) known to be involved in wound healing and chronicity were not found to have significant differences between groups (**c**,**d**); however, there were significant differences in MMP levels themselves, with *MMPs 1–3* being more highly expressed in all patients at week 1 (**e**). *p*-value: * < 0.05; *** < 0.001; **** < 0.0001; ns: not significant. For cytokines *MIG*, *IL1a*, and *ITAC*, *t*-test was used for normally distributed data; all other tests utilized Mann–Whitney U for non-parametric data.

**Figure 5 biomolecules-15-00395-f005:**
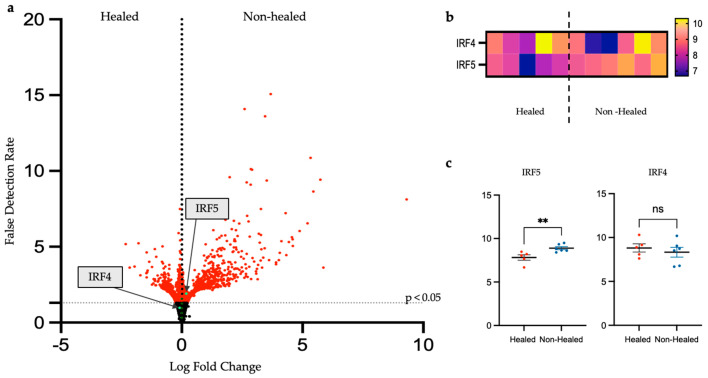
Transcription factors *IRF4* and *IRF5* from debrided tissue at week 1 show *IRF5* having a significant increase in expression in non-healed wound. The expression levels for specific genes *IRF4* and *IRF5* were compared using log2 fold change between groups (**a**). Transcription factor, *IRF5*, which promotes pro-inflammatory (M-1 like) macrophage polarization, was found to have significantly increased expression in the non-healed cohort. While *IRF4*, which promotes anti-inflammatory (M-2 like) macrophage polarization, was not significantly elevated in the healed cohort. (**b**) A heatmap of individual gene expression in all 11 patient samples. (**c**) *IRF5* is significantly elevated in the non-healed cohort, while there is no difference in *IRF4* expression between groups. *p*-value: ** < 0.01, ns: not significant, panel (**c**) significance was calculated using Mann–Whitney U test for non-parametric data.

**Figure 6 biomolecules-15-00395-f006:**
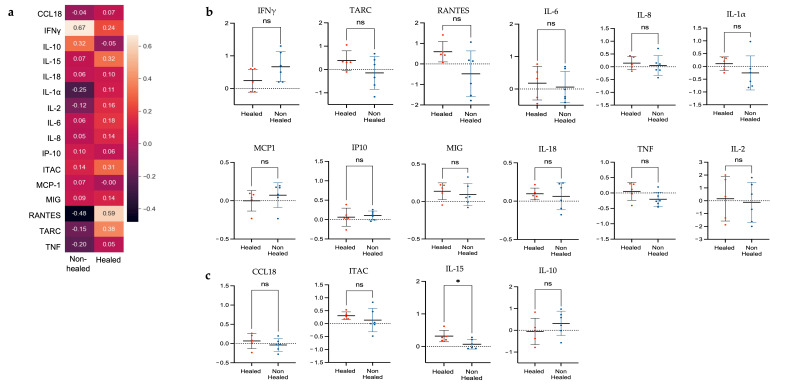
Changes in group average and individual plasma cytokine levels reveal significant increase in IL-15 among healed patients. Plasma cytokines and chemokines were measured at week 1 and week 10 from all patients. The average percent change from week 1 to week 10 levels between groups was calculated to compare trends over time (**a**). Interferon gamma while increased in both groups, showed a 2.8-fold increase in the non-healed cohort compared to the healed cohort. RANTES, showed a 59% increase in healed patients, and a 48% decrease in non-healed patients. The percent change over time was also compared for individual patients in each group (**b**). Panel analyzed using Mann–Whitney U test for non-parametric data. The differences in average percent change for IFNγ and RANTES were found to be non-significant when looking at the variation in individual patients. (**c**). *p*-value: * < 0.05; ns: not significant. Panel analyzed using *t*-test for normally distributed data. Only IL-15 had a significant difference between groups, with a 32% increase by week 10 in the healed cohort, compared to only a 7% increase in non-healed cohort.

**Figure 7 biomolecules-15-00395-f007:**
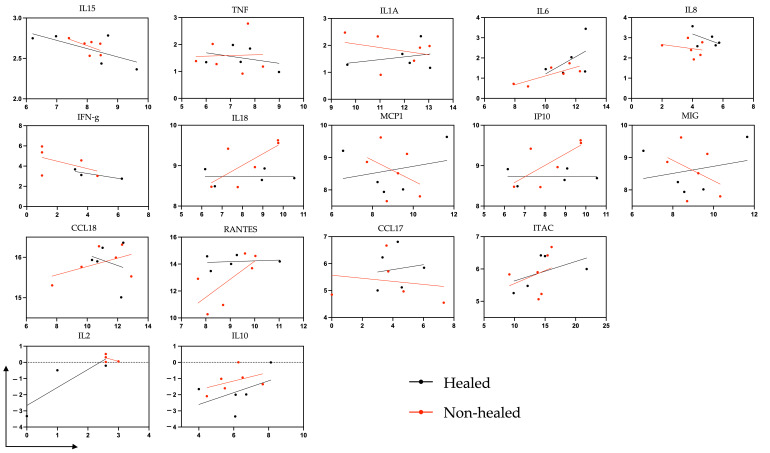
Scatter plot of correlation between mRNA and plasma cytokine levels at week 1. Correlation between gene expression of debrided tissue taken at week 1 and cytokine levels of plasma taken at week 1. Transcription levels were measured using Illumina NextSeq and expression between groups was assessed based on log2 fold changes. Multiplex Elisa was used to measure plasma cytokine levels (pg/mL). All values are shown as log2.

**Figure 8 biomolecules-15-00395-f008:**
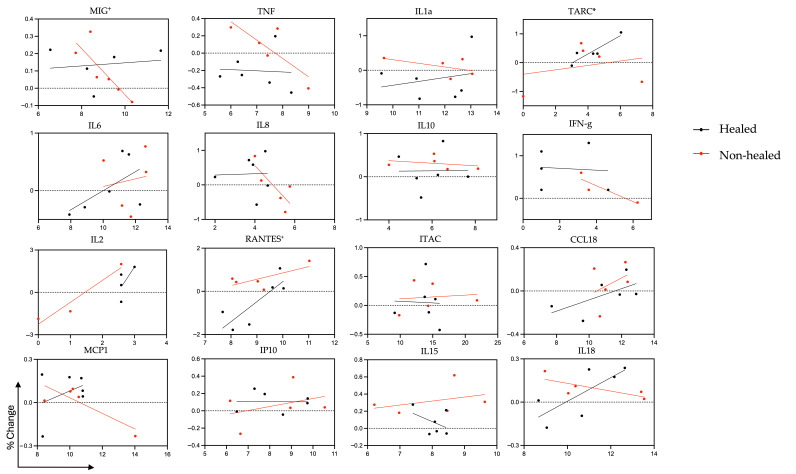
Scatter plot of correlation between mRNA taken at week 1 and plasma cytokine level change over 10 weeks. Correlation between gene expression of debrided tissue taken at week 1 and the percent change in cytokine levels of plasma between week 1 and week 10. Transcription levels were measured using Illumina NextSeq and expression between groups was assessed based on log2 fold changes. Multiplex Elisa was used to measure plasma cytokine levels (pg/mL). The *y*-axis represents the change in plasma cytokine levels between week 10 and week 1. The *x*-axis represents log2 normalized mRNA levels. *p*-value: * < 0.05 in healed; + < 0.05 in non-healed.

**Table 1 biomolecules-15-00395-t001:** Patient demographics and comorbidities.

	Characteristic	Fequency	Percent
Age	40–49	2	18.2
	50–59	3	27.3
	>60	6	54.5
Sex	Male	7	63.6
	Female	4	36.4
Race	White	4	36.4
	African American	4	36.4
	Asian	1	9.1
	Arabic	1	9.1
	Other	1	9.1
Ethnicity	Hispanic	1	9.1
	Non-Hispanic	10	90.9
BMI	18–24	1	9.1
	25–30	5	45.5
	31–40	5	45.5
Smoking Status	Never	4	36.4
	Former	7	63.6
Comorbidities	Hypertension	8	72.7
	Hyperlipidemia	4	36.4
	Coronary Artery Disease	2	18.2
	Pre-Diabetes	2	18.2
	History of VTE	2	18.2
	Lymphedema	2	18.2
	Chronic Venous Insufficiency	11	100

**Table 2 biomolecules-15-00395-t002:** Patient ulcer characteristics.

	Healed	Unhealed	*p*-Value
Ulcer Size (cm^2^)	9.37 (±6.64)	11.2 (±9.80)	0.792
Ulcer Age (months)	7.8 (±4.02)	39.5 (±69.2)	0.365
Healing Time (months)	4.42 (±2.53)	13.7 (±11.0)	0.077

**Table 3 biomolecules-15-00395-t003:** Top 20 protein coding RNA expressed in non-healed wounds at initial presentation.

Pathway	Gene	Log Fold Change	False Detection Rate	Adjusted *p*-Value
Cell Adhesion	WNT1-inducible Signaling Pathway Protein 2 (WISP 2)	2.604	14.08	9.87 × 10^−11^
	Integrin Beta-like 1 Protein (ITGBL1)	3.451	13.6	2.02 × 10^−11^
Cell Signaling, Growth, and Migration	Phosphodiesterase 11A (PDE11A)	5.21	6.53	3.27 × 10^−4^
	Regulator of G-protein Signaling 4 (RGS4)	2.377	6.77	2.26 × 10^−4^
	Phospholipase A2, Group XVI (PLA2G16)	1.808	6.54	3.27 × 10^−4^
	ST6-Beta-Galactoside Alpha 2,6 Sialyltransferase (ST6GAL2)	2.706	7.04	1.29 × 10^−4^
	Complement C1q Tumor Necrosis Factor-related Protein 3 (C1QTNF3)	2.866	10.13	3.28 × 10^−11^
	Cell Growth Regulator with EF-hand Domain 1 (CGREF1)	1.307	6.1	7.67 × 10^−4^
	Sodium Leak Channel (NALCN)	3.521	9.37	1.15 × 10^−6^
Extracellular Matrix	Thrombospondin 4 (THBS4)	3.678	15.07	2.03 × 10^−11^
	Dermatopontin (DPT)	1.982	9.59	8.94 × 10^−7^
	Hyaluronan and Proteoglycan Link Protein 1 (HALPN1)	5.447	8.54	4.95 × 10^−6^
	Fibroblast Growth Factor 14 (FGF14)	2.909	10.09	3.28 × 10^−7^
	RAN Binding Protein 3-like (RANBP3L)	4.302	7.22	9.01 × 10^−5^
Fibrosis and Scarring	Microfibrillar-associated protein 5 (MFAP5)	2.77	6.67	2.55 × 10^−4^
Chondrogenesis and Bone Formation	Osteoglycin (OGN)	3.281	7.5	5.21 × 10^−5^
	Epiphycan (EPYC)	5.734	9.42	1.15 × 10^−6^
	Collagen alpha-1 (XI) (COL11A1)	2.685	9.24	1.37 × 10^−6^
	Stimulator of Chondrogenesis 1 (SCRG1)	4.884	6.04	8.5 × 10^−4^
Neuronal Development	Synapse Differentiation Inducing 1 (SYNDIG1)	2.851231	9.09283	1.77 × 10^−6^

## Data Availability

The raw data supporting the conclusions of this article will be made available by the authors on request.

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
