# Peer review of "Elevation of Plasma IL-15 and RANTES as Potential Biomarkers of Healing in Chronic Venous Ulcerations: A Pilot Study"

_biomolecules, 2025, doi:10.3390/biom15030395_

Round 1
Reviewer 1 Report
Comments and Suggestions for Authors
Thank you for submitting this paper for review. I have the following major and minor comments. This is a translational project exploring the expression of specific biomarkers in healing and non healing chronic venous ulcers.
Major
Methodology:
1. Please provide the rationale for 10 week follow up to assess healing. Many other publications examine ulcer outcomes at 12-20 week follow up.
2. Please provide details regarding the sample size calculation. This is important as you have 11 patients and that is a small number
3. Please provide greater details regarding the participant demographics, metadata, comorbidities, BMI, medications, etc. There may be important confounding variables that have not been mentioned. Please also include the age and size of the ulcers.
4. You mention that the debrided tissue was analysed- which part? The debrided tissue (start / end of debridement, necrotic tissue) or the biopsies from the ulcer bed? Why were no samples taken from the ulcer edge?
Discussion:
1. Which biomarker is most promising? Please describe in greater detail the clinical applications of your research.
Minor
- line 104 - what was your definition of ulcer chronicity? > 2 weeks? > 6 months? Please define
- Line 184 - please provide details on the sample size calculation
- Lines 188 - 197 - please consider a flow diagram to describe your patient flow. This is presented rather confusingly in prose at the moment, and the excluded numbers don’t tally with the remaining patients in the study. Please clarify with a diagram.
- Lines 188 - 197 - as mentioned before, please include patient and ulcer metadata here.
- Line 207 - ‘tissue samples following wound debridement’. Do you mean the punch biopsies or are these samples of debrided tissue? Please clarify.
- Lines 523 - 531 - would you say that your early sampling was a limitation of the study? Please discuss
Line 530 - please expand the clinical implications of the study focussing on the most promising bio markers.
Author Response
Thank you very much for taking the time to review this manuscript. Please find the detailed responses below and the corresponding revisions/corrections highlighted in red in the re-submitted files.
Major
1. Please provide the rationale for 10 week follow up to assess healing. Many other publications examine ulcer outcomes at 12-20 week follow up.
Thank you for your comment. We continued to follow patients beyond 12 weeks if their wounds persisted, however, we stopped collecting samples at 10 weeks due to compliance concerns. This has been included in the section 2.1.2 in red.
2. Please provide details regarding the sample size calculation. This is important as you have 11 patients and that is a small number
We agree that the initial sample size is small. As this was a pilot study, we did not perform a power analysis. Our IRB is approved for 25 patients. However, we calculated a sample size of at least 45 patients for future work.
This is reflected in the Ethics Statement 2.1.1. in red.
3. Please provide greater details regarding the participant demographics, metadata, comorbidities, BMI, medications, etc. There may be important confounding variables that have not been mentioned. Please also include the age and size of the ulcers.
The patient numbers were adjusted, thank you for pointing out this error. A flow diagram was included to supplement the information. We have included a table with the requested patient data in the results section.
4. You mention that the debrided tissue was analyzed- which part? The debrided tissue (start / end of debridement, necrotic tissue) or the biopsies from the ulcer bed? Why were no samples taken from the ulcer edge?
Thank you for pointing out this oversight. In section 2.2.1, we have updated the description of wound tissue collection to reflect better that the debrided tissue was discarded, and two punch biopsies were taken, which included the wound edge.
5. Which biomarker is most promising? Please describe in greater detail the clinical applications of your research.
Thank you for your comments. We have updated the discussion in red to include this analysis.
Minor
We thank the reviewer for their comments. Some of these points were addressed in the major revisions above. If they were not, we have addressed them below in red.
- line 104 - what was your definition of ulcer chronicity? > 2 weeks? > 6 months? Please define
- Line 184 - please provide details on the sample size calculation
Thank you for pointing this out, we have updated Section 2.1.2.
- Lines 188 - 197 - please consider a flow diagram to describe your patient flow. This is presented rather confusingly in prose at the moment, and the excluded numbers don’t tally with the remaining patients in the study. Please clarify with a diagram.
We appreciate the feedback and have included this information as Figure 1.
- Lines 188 - 197 - as mentioned before, please include patient and ulcer metadata here.
We appreciate the feedback and have included this information as Table 1 and have added a paragraph for ulcer sizes/ages.
- Line 207 - ‘tissue samples following wound debridement’. Do you mean the punch biopsies or are these samples of debrided tissue? Please clarify.
Thank you for your comment, we have updated section 2.2.1. to reflect that the biopsies were taken of the wound edge and debrided tissue was discarded.
- Lines 523 - 531 - would you say that your early sampling was a limitation of the study? Please discuss
Yes, we agree that our early sampling is a limitation of this study. We have attempted to make this point more clear in the discussion, which has been updated in red.
Line 530 - please expand the clinical implications of the study focusing on the most promising bio markers.
Thank you for your comments, this has been addressed in red in the discussion.
Reviewer 2 Report
Comments and Suggestions for Authors 1. The authors have written that "Approximately 4mL of whole blood was collected in an EDTA tube and plasma was collected according to standard protocols [19]." Were the patients fasting? Please give this information, even if it is given in the cited document. Please indicate manufacturers of the tubes. 2. The authors claimed "Serum cytokines were measured over time.", but in the previous sentence it was written that blood was drawn into EDTA tubes and plasma was collected. So cytokines were measured in the plasma or in the serum? Additionally chapter 3.3. is titled "Plasma Cytokine Analysis" . Along the whole text serum and plasma are used as synonyms, but they are not. If the serum was used, describe the collection of serum. 3. Selection of measured cytokines shall be justified. 4. The authors used the t-test. Did the authors check the distribution of variables? Was it gaussian? 5. Figure 6 is not readable, at least in embed form. Also the scales of the X axis and Y axis shall be adjusted, which makes charts more informative. 6. Figure 7 - the same comments like for Figure 6. 7. " 3.3.3. Predictive potential of serum biomarkers" look at comment 2 8. Please discuss, if the selected biomarkers can be not only markers, but also targets for therapy e.g. mAbs against certain cytokines are widely available.Author Response
Thank you very much for taking the time to review this manuscript. Please find the detailed responses below and the corresponding revisions/corrections highlighted in red in the re-submitted files.
- The authors have written that "Approximately 4mL of whole blood was collected in an EDTA tube and plasma was collected according to standard protocols [19]." Were the patients fasting? Please give this information, even if it is given in the cited document. Please indicate manufacturers of the tubes.
Thank you for your comments, we have updated this section in red in the new document to reflect that patients were not fasting and included the manufacturer of the EDTA tubes.
- The authors claimed, "Serum cytokines were measured over time.", but in the previous sentence it was written that blood was drawn into EDTA tubes and plasma was collected. So cytokines were measured in the plasma or in the serum? Additionally, chapter 3.3. is titled "Plasma Cytokine Analysis”. Along the whole text serum and plasma are used as synonyms, but they are not. If the serum was used, describe the collection of serum.
Thank you for pointing out this inconsistency. The document has been updated to reflect that plasma was collected for the measurement of cytokines.
- Selection of measured cytokines shall be justified.
Thank you for your comments, we utilized a readily available multiplex kit from Luminex which contained inflammatory cytokines, which are commonly studied. We chose this kit to allow for the analysis of multiple cytokines with a single sample.
- The authors used the t-test. Did the authors check the distribution of variables? Was it gaussian?
Thank you for your comments. We have updated all figures with the results of the appropriate statistical tests based on kurtosis analysis. There were no changes in results after adjusting to non-parametric tests (where warranted).
5 and 6. Figure 6 is not readable, at least in embed form. Also, the scales of the X axis and Y axis shall be adjusted, which makes charts more informative. Figure 7 - the same comments as for Figure 6.
Thank you for your feedback. The figure has been adjusted to have a better resolution and clearer labels. We have also adjusted the scales in both figures to better show the distribution of variables and best fit lines.
- " 3.3.3. Predictive potential of serum biomarkers" look at comment 2
This has been updated. Thank you for pointing out this error.
- Please discuss, if the selected biomarkers can be not only markers, but also targets for therapy e.g. mAbs against certain cytokines are widely available.
Thank you for your comments. We have included discussion on this point in red. We have discussed possible therapeutic potential and challenges for each biomarker.
Round 2
Reviewer 1 Report
Comments and Suggestions for Authors
Thank you for resubmitting this manuscript for review. The authors have addressed the previous comments satisfactorily.
I have the following additional comments
- Page 2 lines 72-76 - you mention conservative management options for venous ulceration but do not include any information on intervention. International guidelines recommend endovenous ablation for these patients. Please rewrite this section providing a more updated management approach for VLU.
- Please clarify whether all the patients in your cohort were treated in exactly the same way - debridement, wound care, compression, etc. Did any of them have surgical intervention?
- Page 3 lines 100-103 - you mention you planned enrolment for 20-25 patients. Why did you stop at 11? Usually the total number of patients planned takes into account dropouts / loss to follow up to ensure that the sample size does not reduce. I appreciate this is a pilot study but you do need to justify why you accepted 11 patients when you wanted to recruit 20-25
- page 3 line 105-6 - please clarify whether all of your patients had a duplex ultrasound confirming the ulceration was venous in origin?
- Page 3 line 111 - please clarify when your follow up point was. Was it 10 weeks or 12 weeks post enrolment? This needs to be clarified throughout the paper as it is very confusing (sometimes 10 weeks mentioned sometimes 12)
- Line 117 - make sure acronymsa re defined at the first use (BMI)
- Page 5 lines 207-208 - you are referencing guidelines from 2006. Please use up to date guidelines (ESVS / AVF / NICE)
- Figure 1 - I would suggest presenting this a little bit like a consort diagram with the excluded patients coming out with an arrow and a box, with the assessed patients, etc under the eligible patients for the study. Please include the week of the final blood sample in the box to clarify when follow up was performed.
- Page 56 lines 223-229 - please include this data in table 1. Please present table 1 with characteristics for both the healed and unhealed group, as these were the groups compared in your study. Please include significance testing with respect to the differences in each group if possible. Please also include ranges / SD
- These were all chronic wounds (> 2 weeks, as per your definition) - please can you clarify when the sampling was performed for these wounds? You mention early sampling - was this early as in immediately after recruitment to the study or is this early in terms of early in the ulcer’s lifetime? You mention that in the non healed groups the approximate wound chronicity was 40 months, over 3 years. In the healed groups it was 8 months.
- On this point, page 16 lines 498 - 501 - you mention the inflammatory gene set was more enriched in the healed cohort and that all samples were taken at initial presentation / the wounds were in the inflammatory phase of healing. If I have understood correctly, you took the samples at recruitment, and the average age of the healed group was still > 6 months, which is well into chronic ulceration territory with all the associated biochemical / mechanistic changes that come with this. This argument would make sense if the wounds were acute. Please clarify.
- Page 18 lines 591-599 - I would suggest stating that this study has a number of limitations. I would also suggest that the presence of multiple wounds is not a minor limitation, rather a very important one, as this may have affected your systemic profile and is a major confounder.
Author Response
Thank you for your insightful comments. Each point has been addressed in the attached PDF and updates to the manuscript are highlighted in red.
